# Cormorant: Covariant Molecular Neural Networks

**Brandon Anderson**[*‡]**, Truong-Son Hy**[*] **and Risi Kondor**[*†‡]

[*]Department of Computer Science, [†]Department of Statistics
The University of Chicago
[♯] Center for Computational Mathematics, Flatiron Institute
[‡] Atomwise
{hytruongson,risi}@uchicago.edu
brandona@jfi.uchicago.edu

## Abstract

We propose *Cormorant*, a rotationally covariant neural network architecture for learning the behavior and properties of complex many-body physical systems. We apply these networks to molecular systems with two goals: learning atomic potential energy surfaces for use in Molecular Dynamics simulations, and learning ground state properties of molecules calculated by Density Functional Theory. Some of the key features of our network are that (a) each neuron explicitly corresponds to a subset of atoms; (b) the activation of each neuron is covariant to rotations, ensuring that overall the network is fully rotationally invariant. Furthermore, the non-linearity in our network is based upon tensor products and the Clebsch-Gordan decomposition, allowing the network to operate entirely in Fourier space. *Cormorant* significantly outperforms competing algorithms in learning molecular Potential Energy Surfaces from conformational geometries in the MD-17 dataset, and is competitive with other methods at learning geometric, energetic, electronic, and thermodynamic properties of molecules on the GDB-9 dataset.

## 1 Introduction

In principle, quantum mechanics provides a perfect description of the forces governing the behavior of atoms, molecules and crystalline materials such as metals. However, for systems larger than a few dozen atoms, solving the Schrödinger equation explicitly at every timestep is not a feasible proposition on present day computers. Even Density Functional Theory (DFT) [Hohenberg and Kohn, 1964], a widely used approximation to the equations of quantum mechanics, has trouble scaling to more than a few hundred atoms.

Consequently, the majority of practical work in molecular dynamics today falls back on fundamentally classical models, where the atoms are essentially treated as solid balls and the forces between them are given by pre-defined formulae called *atomic force fields* or *empirical potentials*, such as the CHARMM family of models [Brooks et al., 1983, 2009]. There has been a widespread realization that this approach has inherent limitations, so in recent years a burgeoning community has formed around trying to use machine learning to *learn* more descriptive force fields directly from DFT computations [Behler and Parrinello, 2007, Bartók et al., 2010, Rupp et al., 2012, Shapeev, 2015, Chmiela et al., 2016, Zhang et al., 2018, Schütt et al., 2017, Hirn et al., 2017]. More broadly, there is considerable interest in using ML methods not just for learning force fields, but also for predicting many other physical/chemical properties of atomic systems across different branches of materials science, chemistry and pharmacology [Montavon et al., 2013, Gilmer et al., 2017, Smith et al., 2017, Yao et al., 2018].

At the same time, there have been significant advances in our understanding of the equivariance and covariance properties of neural networks, starting with [Cohen and Welling, 2016a,b] in the

context of traditional convolutional neural nets (CNNs). Similar ideas underly generalizations of CNNs to manifolds [Masci et al., 2015, Monti et al., 2016, Bronstein et al., 2017] and graphs [Bruna et al., 2014, Henaff et al., 2015]. In the context of CNNs on the sphere, Cohen et al. [2018] realized the advantage of using "Fourier space" activations, i.e., expressing the activations of neurons in a basis defined by the irreducible representations of the underlying symmetry group (see also [Esteves et al., 2017]), and these ideas were later generalized to the entire SE(3) group [Weiler et al., 2018]. Kondor and Trivedi [2018] gave a complete characterization of what operations are allowable in Fourier space neural networks to preserve covariance, and Cohen et al generalized the framework even further to arbitrary gauge fields [Cohen et al., 2019]. There have also been some recent works where even the nonlinear part of the neural network's operation is performed in Fourier space: independently of each other [Thomas et al., 2018] and [Kondor, 2018] were to first to use the Clebsch–Gordan transform inside rotationally covariant neural networks for learning physical systems, while [Kondor et al., 2018] showed that in spherical CNNs the Clebsch–Gordan transform is sufficient to serve as the sole source of nonlinearity.

The *Cormorant* neural network architecture proposed in the present paper combines some of the insights gained from the various force field and potential learning efforts with the emerging theory of Fourier space covariant/equivariant neural networks. The important point that we stress in the following pages is that by setting up the network in such a way that each neuron corresponds to an actual set of physical atoms, and that each activation is covariant to symmetries (rotation and translation), we get a network in which the "laws" that individual neurons learn resemble known physical interactions. Our experiments show that this generality pays off in terms of performance on standard benchmark datasets.

## 2 The nature of physical interactions in molecules

Ultimately interactions in molecular systems arise from the quantum structure of electron clouds around constituent atoms. However, from a chemical point of view, effective atom-atom interactions break down into a few simple classes based upon symmetry. Here we review a few of these classes in the context of the multipole expansion, whose structure will inform the design of our neural network.

**Scalar interactions.** The simplest type of physical interaction is that between two particles that are pointlike and have no internal directional degrees of freedom, such as spin or dipole moments. A classical example is the electrostatic attraction/repulsion between two charges described by the Coulomb energy

$$V_C = -\frac{1}{4\pi\epsilon_0} \frac{q_A q_B}{|\boldsymbol{r}_{AB}|} . \tag{1}$$

Here $q_A$ and $q_B$ are the charges of the two particles, $\boldsymbol{r}_A$ and $\boldsymbol{r}_B$ are their position vectors, $\boldsymbol{r}_{AB} = \boldsymbol{r}_A - \boldsymbol{r}_B$, and $\epsilon_0$ is a universal constant. Note that this equation already reflects symmetries: the fact that (1) only depends on the *length* of $\boldsymbol{r}_{AB}$ and not its direction or the position vectors individually guarantees that the potential is invariant under both translations and rotations.

**Dipole/dipole interactions.** One step up from the scalar case is the interaction between two dipoles. In general, the electrostatic dipole moment of a set of $N$ charged particles relative to their center of mass $\boldsymbol{r}$ is just the first moment of their position vectors weighted by their charges:

$$\boldsymbol{\mu} = \sum_{i=1}^{N} q_i (\boldsymbol{r}_i - \boldsymbol{r}).$$

The dipole/dipole contribution to the electrostatic potential energy between two sets of particles $A$ and $B$ separated by a vector $\boldsymbol{r}_{AB}$ is then given by

$$V_{d/d} = \frac{1}{4\pi\epsilon_0} \left[ \frac{\boldsymbol{\mu}_A \cdot \boldsymbol{\mu}_B}{|\boldsymbol{r}_{AB}|^3} - 3 \frac{(\boldsymbol{\mu}_A \cdot \boldsymbol{r}_{AB})(\boldsymbol{\mu}_B \cdot \boldsymbol{r}_{AB})}{|\boldsymbol{r}_{AB}|^5} \right]. \tag{2}$$

One reason why dipole/dipole interactions are indispensible for capturing the energetics of molecules is that most chemical bonds are polarized. However, dipole/dipole interactions also occur in other contexts, such as the interaction between the magnetic spins of electrons.

**Quadropole/quadropole interactions.** One more step up the multipole hierarchy is the interaction between quadropole moments. In the electrostatic case, the quadropole moment is the second moment of the charge density (corrected to remove the trace), described by the *matrix*

$$\boldsymbol{\Theta} = \sum_{i=1}^{N} q_i (3\boldsymbol{r}_i \boldsymbol{r}_i^\top - |\boldsymbol{r}_i|^2 I).$$

Quadropole/quadropole interactions appear for example when describing the interaction between benzene rings, but the general formula for the corresponding potential is quite complicated. As a simplification, let us only consider the special case when in some coordinate system aligned with the structure of $A$, and at polar angle $(\theta_A, \phi_A)$ relative to the vector $\boldsymbol{r}_{AB}$ connecting $A$ and $B$, $\boldsymbol{\Theta}_A$ can be transformed into a form that is diagonal, with $[\Theta_A]_{zz} = \vartheta_A$ and $[\Theta_A]_{xx} = [\Theta_A]_{yy} = -\vartheta_A/2$ [Stone, 1997]. We make a similar assumption about the quadropole moment of $B$. In this case the interaction energy becomes

$$V_{q/q} = \frac{3}{4} \frac{\vartheta_A \vartheta_B}{4\pi\epsilon_0 |\boldsymbol{r}_{AB}|^5} \big[ 1 - 5\cos_A^\theta - 5\cos^2\theta_B - 15\cos^2\theta_A \cos^2\theta_B +$$

$$2(4\cos\theta_A\theta_B - \sin\theta_A \sin\theta_B \cos(\phi_A - \phi_B))^2 \big]. \quad (3)$$

Higher order interactions involve moment tensors of order 3,4,5, and so on. One can appreciate that the corresponding formulae, especially when considering not just electrostatics but other types of interactions as well (dispersion, exchange interaction, etc), quickly become very involved.

## 3  Spherical tensors and representation theory

Fortunately, there is an alternative formalism for expressing molecular interactions, that of spherical tensors, which makes the general form of physically allowable interactions more transparent. This formalism also forms the basis of the our Cormorant networks described in the next section.

The key to spherical tensors is understanding how physical quantities transform under rotations. Specifically, in our case, under a rotation $\boldsymbol{R}$:

$$q \longmapsto q \qquad \boldsymbol{\mu} \longmapsto \boldsymbol{R}\boldsymbol{\mu} \qquad \boldsymbol{\Theta} \longmapsto \boldsymbol{R}\boldsymbol{\Theta}\boldsymbol{R}^\top \qquad \boldsymbol{r}_{AB} \longmapsto \boldsymbol{R}\boldsymbol{r}_{AB}.$$

Flattening $\boldsymbol{\Theta}$ into a vector $\overline{\boldsymbol{\Theta}} \in \mathbb{R}^9$, its transformation rule can equivalently be written as $\overline{\boldsymbol{\Theta}} \mapsto (\boldsymbol{R} \otimes \boldsymbol{R})\overline{\boldsymbol{\Theta}}$, showing its similarity to the other three cases. In general, a $k$'th order Cartesian moment tensor $T^{(k)} \in \mathbb{R}^{3 \times 3 \times \ldots \times 3}$ (or its flattened $\overline{T}^{(k)} \in \mathbb{R}^{3k}$ equivalent) transforms as $\overline{T}^{(k)} \mapsto (\boldsymbol{R} \otimes \boldsymbol{R} \otimes \ldots \otimes \boldsymbol{R})\overline{T}^{(k)}$.

Recall that given a group $G$, a *representation* $\rho$ of $G$ is a matrix valued function $\rho\colon G \to \mathbb{C}^{d \times d}$ obeying $\rho(xy) = \rho(x)\rho(y)$ for any two group elements $x, y \in G$. It is easy to see that $\boldsymbol{R}$, and consequently $\boldsymbol{R} \otimes \ldots \otimes \boldsymbol{R}$ are representations of the three dimensional rotation group SO(3). We also know that because SO(3) is a compact group, it has a countable sequence of unitary so-called irreducible representations (irreps), and, up to a similarity transformation, any representation can be reduced to a direct sum of irreps. In the specific case of SO(3), the irreps are called *Wigner D-matrices* and for any positive integer $\ell = 0, 1, 2, \ldots$ there is a single corresponding irrep $D^\ell(\boldsymbol{R})$, which is a $(2\ell + 1)$ dimensional representation (i.e., as a function, $D^\ell\colon \text{SO}(3) \to \mathbb{C}^{(2\ell+1) \times (2\ell+1)}$). The $\ell = 0$ irrep is the trivial irrep $D^0(\boldsymbol{R}) = (1)$.

The above imply that there is a fixed unitary transformation matrix $C^{(k)}$ which reduces the $k$'th order rotation operator into a direct sum of irreducible representations:

$$\underbrace{\boldsymbol{R} \otimes \boldsymbol{R} \otimes \ldots \otimes \boldsymbol{R}}_{k} = C^{(k)} \Big[ \bigoplus_\ell \bigoplus_{i=1}^{\tau_\ell} D^\ell(\boldsymbol{R}) \Big] C^{(k)\dagger}.$$

Note that the transformation $\boldsymbol{R} \otimes \boldsymbol{R} \otimes \ldots \otimes \boldsymbol{R}$ contains redundant copies of $D^\ell(\boldsymbol{R})$, which we denote as the multiplicites $\tau_\ell$. For our present purposes knowing the actual values of the $\tau_\ell$ is not that important, except that $\tau_k = 1$ and that for any $\ell > k$, $\tau_\ell = 0$. What is important is that $\overline{T}^{(k)}$, the vectorized form of the Cartesian moment tensor has a corresponding decomposition

$$\overline{T}^{(k)} = C^{(k)} \Big[ \bigoplus_\ell \bigoplus_{i=1}^{\tau_\ell} Q_{\ell,i} \Big]. \quad (4)$$

This is nice, because using the unitarity of $Q_{\ell,i}$, it shows that under rotations the individual $Q_{\ell,i}$ components transform *independently* as $Q_{\ell,i} \mapsto D^\ell(\boldsymbol{R})\, Q_{\ell,i}$.

What we have just described is a form of generalized Fourier analysis applied to the transformation of Cartesian tensors under rotations. For the electrostatic multipole problem it is particularly relevant, because it turns out that in that case, due to symmetries of $\overline{T}^{(k)}$, the only nonzero $Q_{\ell,i}$ component of (4) is the single one with $\ell = k$. Furthermore, for a set of $N$ charged particles (indexing its components $-\ell, \ldots, \ell$) $Q_\ell$ has the simple form

$$[Q_\ell]_m = \left( \frac{4\pi}{2\ell+1} \right)^{1/2} \sum_{i=1}^{N} q_i\, (r_i)^\ell\, Y_\ell^m(\theta_i, \phi_i) \qquad m = -\ell, \ldots, \ell, \qquad (5)$$

where $(r_i, \theta_i, \phi_i)$ are the coordinates of the $i$'th particle in spherical polars, and the $Y_\ell^m(\theta, \phi)$ are the well known spherical harmonic functions. $Q_\ell$ is called the $\ell$'th *spherical moment* of the charge distribution. Note that while $\overline{T}^{(\ell)}$ and $Q_\ell$ convey exactly the same information, $\overline{T}^{(\ell)}$ is a tensor with $3^\ell$ components, while $Q_\ell$ is just a $(2\ell+1)$ dimensional vector.

Somewhat confusingly, in physics and chemistry any quantity $U$ that transforms under rotations as $U \mapsto D^\ell(\boldsymbol{R})U$ is often called an ($\ell$'th order) *spherical tensor*, despite the fact that in terms of its presentation $Q_\ell$ is just a vector of $2\ell+1$ numbers. Also note that since $D^0(\boldsymbol{R}) = (1)$, a zeroth order spherical tensor is just a scalar. A first order spherical tensor, on the other hand, can be used to represent a spatial vector $\boldsymbol{r} = (r, \theta, \phi)$ by setting $[U_1]_m = r\, Y_1^m(\theta, \phi)$.

## 3.1 The general form of interactions

The benefit of the spherical tensor formalism is that it makes it very clear how each part of a given physical equation transforms under rotations. For example, if $Q_\ell$ and $\widetilde{Q}_\ell$ are two $\ell$'th order spherical tensors, then $Q_\ell^\dagger \widetilde{Q}_\ell$ is a scalar, since under a rotation $\boldsymbol{R}$, by the unitarity of the Wigner $D$-matrices,

$$Q_\ell^\dagger \widetilde{Q}_\ell \longmapsto (D^\ell(\boldsymbol{R})\, Q_\ell)^\dagger\, (D^\ell(\boldsymbol{R})\, \widetilde{Q}_\ell) = Q_\ell^\dagger\, (D^\ell(\boldsymbol{R}))^\dagger\, D^\ell(\boldsymbol{R})\, \widetilde{Q}_\ell = Q_\ell^\dagger \widetilde{Q}_\ell.$$

Even the dipole/dipole interaction (2) requires a more sophisticated way of coupling spherical tensors than this, since it involves non-trivial interactions between not just two, but three different quantites: the two dipole moments $\boldsymbol{\mu}_A$ and $\boldsymbol{\mu}_B$ and the the relative position vector $\boldsymbol{r}_{AB}$. Representing interactions of this type requires taking *tensor products* of the constituent variables. For example, in the dipole/dipole case we need terms of the form $Q_{\ell_1}^A \otimes Q_{\ell_2}^B$. Naturally, these will transform according to the tensor product of the corresponding irreps:

$$Q_{\ell_1}^A \otimes Q_{\ell_2}^B \mapsto (D^{\ell_1}(\boldsymbol{R}) \otimes D^{\ell_2}(\boldsymbol{R}))\, (Q_{\ell_1}^A \otimes Q_{\ell_2}^B).$$

In general, $D^{\ell_1}(\boldsymbol{R}) \otimes D^{\ell_2}(\boldsymbol{R})$ is *not* an irreducible representation. However it does have a well studied decomposition into irreducibles, called the *Clebsch–Gordan* decomposition:

$$D^{\ell_1}(\boldsymbol{R}) \otimes D^{\ell_2}(\boldsymbol{R}) = C_{\ell_1,\ell_2}^\dagger \left[ \bigoplus_{\ell=|\ell_1-\ell_2|}^{\ell_1+\ell_2} D^\ell(\boldsymbol{R}) \right] C_{\ell_1,\ell_2}.$$

Letting $C_{\ell_1,\ell_2,\ell} \in \mathbb{C}^{(2\ell+1) \times (2\ell_1+1)(2\ell_2+2)}$ be the block of $2\ell+1$ rows in $C_{\ell_1,\ell_2}$ corresponding to the $\ell$ component of the direct sum, we see that $C_{\ell_1,\ell_2,\ell}(Q_{\ell_1}^A \otimes Q_{\ell_2}^B)$ is an $\ell$'th order spherical tensor. In particular, given some other spherical tensor quantity $U_\ell$,

$$U_\ell^\dagger \cdot C_{\ell_1,\ell_2,\ell} \cdot (Q_{\ell_1}^A \otimes Q_{\ell_2}^B)$$

is a scalar, and hence it is a candidate for being a term in the potential energy. Note the similarity of this expression to the *bispectrum* [Kakarala, 1992, Bendory et al., 2018], which is an already established tool in the force field learning literature [Bartók et al., 2013].

Almost any rotation invariant interaction potential can be expressed in terms of iterated Clebsch–Gordan products between spherical tensors. In particular, the full electrostatic energy between two sets of charges $A$ and $B$ separated by a vector $\boldsymbol{r} = (r, \theta, \phi)$ expressed in multipole form [Jackson, 1999] is

$$V_{AB} = \frac{1}{4\pi\epsilon_0} \sum_{\ell=0}^{\infty} \sum_{\ell'=0}^{\infty} \sqrt{\binom{2\ell+2\ell'}{2\ell}} \sqrt{\frac{4\pi}{2\ell+2\ell'+1}}\, r^{-(\ell+\ell'+1)}\, Y_{\ell+\ell'}(\theta, \phi)\, C_{\ell_1,\ell_2,\ell+\ell'}\, (Q_\ell^A \otimes Q_{\ell'}^B).$$

$$(6)$$

Note the generality of this formula: the $\ell = \ell' = 1$ case covers the dipole/dipole interaction (2), the $\ell = \ell' = 2$ case covers the quadropole/quadropole interaction (3), while the other terms cover every other possible type of multipole/multipole interaction. Magnetic and other types of interactions, including interactions that involve 3-way or higher order terms, can also be recovered from appropriate combinations of tensor products and Clebsch–Gordan decompositions.

We emphasize that our discussion of electrostatics is only intended to illustrate the algebraic structure of interatomic interactions of any type, and is not restricted to electrostatics. In what follows, we will not explicitly specify what interactions the network will learn. Nevertheless, there are physical constraints on the interactions arising from symmetries, which we explicitly impose in our design of Cormorant.

# 4 CORMORANT: COvaRiant MOleculaR Artificial Neural neTworks

The goal of using ML in molecular problems is not to encode known physical laws, but to provide a platform for learning interactions from data that cannot easily be captured in a simple formula. Nonetheless, the mathematical structure of known physical laws, like those discussed in the previous sections, give strong hints about how to represent physical interactions in algorithms. In particular, when using machine learning to learn molecular potentials or similar rotation and translation invariant physical quantities, it is essential to make sure that the algorithm respects these invariances.

Our Cormorant neural network has invariance to rotations baked into its architecture in a way that is similar to the physical equations of the previous section: the internal activations are all spherical tensors, which are then combined at the top of the network in such a way as to guarantee that the final output is a scalar (i.e., is invariant). However, to allow the network to learn interactions that are more complicated than classical interatomic forces, we allow each neuron to output not just a single spherical tensor, but a combination of spherical tensors of different orders. We will call an object consisting of $\tau_0$ scalar components, $\tau_1$ components transforming as first order spherical tensors, $\tau_2$ components transforming as second order spherical tensors, and so on, an SO(3)–*covariant vector of type* $(\tau_0, \tau_1, \tau_2, \ldots)$. The output of each neuron in Cormorant is an SO(3)–vector of a fixed type.

**Definition 1.** *We say that $F$ is an* SO(3)*-covariant vector of type* $\boldsymbol{\tau} = (\tau_0, \tau_1, \tau_2, \ldots, \tau_L)$ *if it can be written as a collection of complex matrices $F_0, F_1, \ldots, F_L$, called its* isotypic parts, *where each $F_\ell$ is a matrix of size $(2\ell + 1) \times \tau_\ell$ and transforms under rotations as $F_\ell \mapsto D^\ell(\boldsymbol{R}) F_\ell$.*

The second important feature of our architecture is that each neuron corresponds to either a single atom or a set of atoms forming a physically meaningful subset of the system at hand, for example all atoms in a ball of a given radius. This condition helps encourage the network to learn physically meaningful and interpretable interactions. The high level definition of *Cormorant* nets is as follows.

**Definition 2.** *Let $\mathcal{S}$ be a molecule or other physical system consisting of $N$ atoms. A "Cormorant" covariant molecular neural network for $\mathcal{S}$ is a feed forward neural network consisting of $m$ neurons $\mathfrak{n}_1, \ldots, \mathfrak{n}_m$, such that*

*C1. Every neuron $\mathfrak{n}_i$ corresponds to some subset $\mathcal{S}_i$ of the atoms. In particular, each input neuron corresponds to a single atom. Each output neuron corresponds to the entire system $\mathcal{S}$.*

*C2. The activation of each $\mathfrak{n}_i$ is an SO(3)-vector of a fixed type $\boldsymbol{\tau}_i$.*

*C3. The type of each output neuron is $\boldsymbol{\tau}_{out} = (1)$, i.e., a scalar.* [1]

Condition (C3) guarantees that whatever function a *Cormorant* network learns will be invariant to global rotations. Translation invariance is easier to enforce simply by making sure that the interactions represented by individual neurons only involve relative distances.

## 4.1 Covariant neurons

The neurons in our network must be such that if each of their inputs is an SO(3)–covariant vector then so is their output. Classically, neurons perform a simple linear operation such as $\mathbf{x} \mapsto W\mathbf{x} + \boldsymbol{b}$, followed by a nonlinearity like a ReLU. In convolutional neural nets the weights are tied together in

a specific way which guarantees that the activation of each layer is covariant to the action of global translations. Kondor and Trivedi [2018] discuss the generalization of convolution to the action of compact groups (such as, in our case, rotations) and prove that the only possible *linear* operation that is covariant with the group action, is what, in terms of SO(3)–vectors, corresponds to multiplying each $F_\ell$ matrix from the right by some matrix $W$ of learnable weights.

For the nonlinearity, one option would be to express each spherical tensor as a function on SO(3) using an inverse SO(3) Fourier transform, apply a pointwise nonlinearity, and then transform the resulting function back into spherical tensors. This is the approach taken in e.g., [Cohen et al., 2018]. However, in our case this would be forbiddingly costly, as well as introducing quadrature errors by virtue of having to interpolate on the group, ultimately degrading the network's covariance. Instead, taking yet another hint from the structure of physical interactions, we use the Clebsch–Gordan transform introduced in 3.1 as a nonlinearity. The general rule for taking the CG product of two SO(3)–parts $F_{\ell_1} \in \mathbb{C}^{(2\ell_1+1)\times n_1}$ and $G_{\ell_2} \in \mathbb{C}^{(2\ell_2+1)\times n_2}$ gives a collection of parts $[F_{\ell_1} \otimes_{\mathrm{cg}} G_{\ell_2}]_{|\ell_1-\ell_1|}, \dots [F_{\ell_1} \otimes_{\mathrm{cg}} G_{\ell_2}]_{\ell_1+\ell_1}$ with columns

$$\left[[F_{\ell_1} \otimes_{\mathrm{cg}} G_{\ell_2}]_\ell\right]_{*,(i_1,i_2)} = C_{\ell_1,\ell_2,\ell}\left([F_{\ell_1}]_{*,i_1} \otimes [G_{\ell_2}]_{*,i_2}\right), \qquad (7)$$

i.e., every column of $F_{\ell_1}$ is separately CG-multiplied with every column of $G_{\ell_2}$. The $\ell$'th part of the CG-product of two SO(3)–vectors consists of the concatenation of all SO(3)–part matrices with index $\ell$ coming from multiplying each part of $F$ with each part of $G$:

$$[F \otimes_{\mathrm{cg}} G]_\ell = \bigoplus_{\ell_1} \bigoplus_{\ell_2} [F_{\ell_1} \otimes_{\mathrm{cg}} G_{\ell_2}]_\ell.$$

Here and in the following $\oplus$ denotes the appropriate concatenation of vectors and matrices. In Cormorant, however, as a slight departure from (7), to reduce the quadratic blow-up in the number of columns, we always have $n_1 = n_2$ and use the restricted "channel-wise" CG-product,

$$\left[[F_{\ell_1} \otimes_{\mathrm{cg}} G_{\ell_2}]_\ell\right]_{*,i} = C_{\ell_1,\ell_2,\ell}\left([F_{\ell_1}]_{*,i} \otimes [G_{\ell_2}]_{*,i}\right),$$

where each column of $F_{\ell_1}$ is only mixed with the corresponding column of $G_{\ell_2}$. We note that similar Clebsch–Gordan nonlinearities were used in [Kondor et al., 2018], and that the Clebsch–Gordan product is also an essential part of Tensor Field Networks [Thomas et al., 2018].

### 4.2   One-body and two-body interactions

As stated in Definition 2, the covariant neurons in a Cormorant net correspond to different subsets of the atoms making up the physical system to be modeled. For simplicty in our present architecture there are only two types of neurons: those that correspond to individual atoms and those that correspond to pairs. For a molecule consisting of $N$ atoms, each layer $s = 0, 1, \dots, S$ of the covariant part of the network has $N$ neurons corresponding to the atoms and $N^2$ neurons corresponding to the $(i, j)$ atom pairs. By loose analogy with graph neural networks, we call the corresponding $F_i^s$ and $g_{i,j}^s$ activations vertex and edge activations, respectively.

In accordance with the foregoing, each $F_i^s$ activation is an SO(3)–vector consisting of $L+1$ distinct parts $(F_i^{s,0}, F_i^{s,1}, \dots, F_i^{s,L})$, i.e., each $F_i^{s,\ell}$ is a $(2\ell+1)\times\tau_\ell^s$ dimensional complex matrix that transforms under rotations as $F_i^{s,\ell} \mapsto D^\ell(R)\, F_i^{s,\ell}$. The different columns of these matrices are regarded as the different *channels* of the network, because they fulfill a similar role to channels in conventional convolutional nets. The $g_{i,j}^s$ edge activations also break down into parts $(g_{i,j}^{s,0}, g_{i,j}^{s,1}, \dots, g_{i,j}^{s,L})$, but these are invariant under rotations. Again for simplicity, in the version of Cormorant that we used in our experiments $L$ is the same in every layer (specifically $L = 3$), and the number of channels is also independent of both $s$ and $\ell$, specifically, $\tau_\ell^s \equiv n_c = 16$.

The actual form of the vertex activations captures "one-body interactions" propagating information from the previous layer related to the *same* atom and (indirectly, via the edge activations) "two-body interactions" capturing interactions between *pairs* of atoms:

$$F_i^{s-1} = \left[ \underbrace{F_i^s \oplus \left(F_i^{s-1} \otimes_{\mathrm{cg}} F_i^{s-1}\right)}_{\text{one-body part}} \oplus \underbrace{\left(\sum_j G_{i,j}^s \otimes_{\mathrm{cg}} F_j^{s-1}\right)}_{\text{two-body part}} \right] \cdot W_{s,\ell}^{\text{vertex}}. \qquad (8)$$

Here $G_{i,j}^s$ are SO(3)–vectors arising from the edge network. Specifically, $G_{i,j}^{s,\ell} = g_{i,j}^{s,\ell} Y^\ell(\widehat{\boldsymbol{r}}_{i,j})$, where $Y^\ell(\widehat{\boldsymbol{r}}_{i,j})$ are the spherical harmonic vectors capturing the relative position of atoms $i$ and $j$. The edge activations, in turn, are defined

$$g_{i,j}^{s,\ell} = \mu^s(r_{i,j}) \left[ \left( g_{i,j}^{s-1,\ell} \oplus \left( F_i^{s-1} \cdot F_j^{s-1} \right) \oplus \eta^{s,\ell}(r_{i,j}) \right) W_{s,\ell}^{\text{edge}} \right] \qquad (9)$$

where we made the $\ell = 0, 1, \ldots, L$ irrep index explicit. As before, in these formulae, $\oplus$ denotes concatenation over the channel index $c$, $\eta_c^{s,\ell}(r_{i,j})$ are learnable radial functions, and $\mu_c^s(r_{i,j})$ are learnable cutoff functions limiting the influence of atoms that are farther away from atom $i$. The learnable parameters of the network are the $\{W_{s,\ell}^{\text{vertex}}\}$ and $\{W_{s,\ell}^{\text{edge}}\}$ weight matrices.

Note that the $F_i^{s-1} \cdot F_j^{s-1}$ dot product term is the only term in these formulae responsible for the interaction between different atoms, and that this term always appears in conjunction with the $\eta_c^{s,\ell}(r_{i,j})$ radial basis functions and $\mu_c^s(r_{i,j})$ cutoff functions (as well as the SO(3)–covariant spherical harmonic vector) making sure that interaction scales with the distance between the atoms. More details of these activation rules are given in the Supplement.

### 4.3 Overall structure and comparison with other architectures

In addition to the covariant neurons described above, our network also needs neurons to compute the input featurization and the the final output after the covariant layers. Thus, in total, a Cormorant networks consists of three distinct parts:

1. An input featurization network $\{F_j^{s=0}\} \leftarrow \text{INPUT}(\{Z_i, r_{i,j}\})$ that operates only on atomic charges/identities and (optionally) a scalar function of relative positions $r_{i,j}$.
2. An $S$-layer network $\{F_i^{s+1}\} \leftarrow \text{CGNet}(\{F_i^s\})$ of covariant activations $F_i^s$, each of which is a SO(3)-vector of type $\tau_i^s$.
3. A rotation invariant network at the top $y \leftarrow \text{OUTPUT}(\bigoplus_{s=0}^{S} \{F_i^s\})$ that constructs scalars from the activations $F_i^s$, and uses them to predict a regression target $y$.

We leave the details of the input and output featurization to the Supplement.

A key difference between Cormorant and other recent covariant networks (Tensor Field Networks [Thomas et al., 2018] and SE(3)-equivariant networks [Weiler et al., 2018]) is the use of Clebsch-Gordan non-linearities. The Clebsch-Gordan non-linearity results in a complete interaction of every degree of freedom in an activation. This comes at the cost of increased difficulty in training, as discussed in the Supplement. We further note that SE(3)-equivariant networks use a three-dimensional grid of points to represent data, and ensure both translational and rotational covariance (equivariance) of each layer. Cormorant on the other hand uses activations that are covariant to rotations, and strictly invariant to translations.

## 5 Experiments

We present experimental results on two datasets of interest to the computational chemistry community: MD-17 for learning molecular force fields and potential energy surfaces, and QM-9 for learning the ground state properties of a set of molecules. The supplement provides a detailed summary of all hyperparameters, our training algorithm, and the details of the input/output levels used in both cases. Our code is available at https://github.com/risilab/cormorant.

**QM9** [Ramakrishnan et al., 2014] is a dataset of approximately 134k small organic molecules containing the atoms H, C, N, O, F. For each molecule, the ground state configuration is calculated using DFT, along with a variety of molecular properties. We use the ground state configuration as the input to our Cormorant, and use a common subset of properties in the literature as regression targets. Table 1(a) presents our results averaged over three training runs compared with SchNet [Schütt et al., 2017], MPNNs [Gilmer et al., 2017], and wavelet scattering networks [Hirn et al., 2017]. Of the twelve regression targets considered, we achieve leading or competitive results on six ($\alpha$, $\Delta\epsilon$, $\epsilon_{\text{HOMO}}$, $\epsilon_{\text{LUMO}}$, $\mu$, $C_v$). The remaining four targets are within $40\%$ of the best result, with the exception of $R^2$.

**MD-17** [Chmiela et al., 2016] is a dataset of eight small organic molecules (see Table 1(b)) containing up to 17 total atoms composed of the atoms H, C, N, O, F. For each molecule, an *ab*

Table 1: Mean absolute error of various prediction targets on QM-9 (left) and conformational energies (in units of kcal/mol) on MD-17 (right). The best results within a standard deviation of three Cormorant training runs (in parenthesis) are indicated in bold.

| | Cormorant | | SchNet | NMP | WaveScatt |
|---|---|---|---|---|---|
| $\alpha$ (bohr$^3$) | **0.085** | (0.001) | 0.235 | 0.092 | 0.160 |
| $\Delta\epsilon$ (eV) | **0.061** | (0.005) | **0.063** | 0.069 | 0.118 |
| $\epsilon_{\mathrm{HOMO}}$ (eV) | **0.034** | (0.002) | 0.041 | 0.043 | 0.085 |
| $\epsilon_{\mathrm{LUMO}}$ (eV) | **0.038** | (0.008) | **0.034** | **0.038** | 0.076 |
| $\mu$ (D) | **0.038** | (0.009) | 0.033 | **0.030** | 0.340 |
| $C_v$ (cal/mol K) | **0.026** | (0.000) | 0.033 | 0.040 | 0.049 |
| $G$ (eV) | 0.020 | (0.000) | **0.014** | 0.019 | 0.022 |
| $H$ (eV) | 0.021 | (0.001) | **0.014** | 0.017 | 0.022 |
| $R^2$ (bohr$^2$) | 0.961 | (0.019) | **0.073** | 0.180 | 0.410 |
| $U$ (eV) | 0.021 | (0.000) | **0.019** | 0.020 | 0.022 |
| $U_0$ (eV) | 0.022 | (0.003) | **0.014** | 0.020 | 0.022 |
| ZPVE (meV) | 2.027 | (0.042) | 1.700 | **1.500** | 2.000 |

| | Cormorant | DeepMD | DTNN | SchNet | GDML | sGDML |
|---|---|---|---|---|---|---|
| Aspirin | **0.098** | 0.201 | – | 0.120 | 0.270 | 0.190 |
| Benzene | **0.023** | 0.065 | 0.040 | 0.070 | 0.070 | 0.100 |
| Ethanol | **0.027** | 0.055 | – | 0.050 | 0.150 | 0.070 |
| Malonaldehyde | **0.041** | 0.092 | 0.190 | 0.080 | 0.160 | 0.100 |
| Naphthalene | **0.029** | 0.095 | – | 0.110 | 0.120 | 0.120 |
| Salicylic Acid | **0.066** | 0.106 | 0.410 | 0.100 | 0.120 | 0.120 |
| Toluene | **0.034** | 0.085 | 0.180 | 0.090 | 0.120 | 0.100 |
| Uracil | **0.023** | 0.085 | – | 0.100 | 0.110 | 0.110 |

*initio* molecular dynamics simulation was run using DFT to calculate the ground state energy and forces. At intermittent timesteps, the energy, forces, and configuration (positions of each atom) were recorded. For each molecule we use a train/validation/test split of 50k/10k/10k atoms respectively. The results of these experiments are presented in Table 1(b), where the mean-average error (MAE) is plotted on the test set for each of molecules. (All units are in kcal/mol, as consistent with the dataset and the literature.) To the best of our knowledge, the current state-of-the art algorithms on this dataset are DeepMD [Zhang et al., 2018], DTNN [Schütt et al., 2017], SchNet [Schütt et al., 2017], GDML [Chmiela et al., 2016], and sGDML [Chmiela et al., 2018]. Since training and testing set sizes were not consistent, we used a training set of 50k molecules to compare with all neural network based approaches. As can be seen from the table, our *Cormorant* network outperforms all competitors.

# 6 Conclusions

To the best of our knowledge, *Cormorant* is the first neural network architecture in which the operations implemented by the neurons is directly motivated by the form of known physical interactions. Rotation and translation invariance are explicitly "baked into" the network by the fact all activations are represented in spherical tensor form (SO(3)–vectors), and the neurons combine Clebsch–Gordan products, concatenation of parts and mixing with learnable weights, all of which are covariant operations. In future work we envisage the potentials learned by *Cormorant* to be directly integrated in MD simulation frameworks. In this regard, it is very encouraging that on MD-17, which is the standard benchmark for force field learning, *Cormorant* outperforms all other competing methods. Learning from derivatives (forces) and generalizing to other compact symmetry groups are natural extensions of the persent work.

### Acknowledgements

This project was supported by DARPA "Physics of AI" grant number HR0011837139, and used computational resources acquired through NSF MRI 1828629.

## Footnotes

[1]Cormorant can learn data of arbitrary SO(3)-vector outputs. We restrict to scalars here to simplify the exposition.

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
