[Supplementary Material]

# Cormorant: Covariant Molecular Neural Networks Supplemental Material

**Brandon Anderson**[*‡]**, Truong-Son Hy**[*] **and Risi Kondor**[*†‡]
[*]Department of Computer Science, [†]Department of Statistics
The University of Chicago
[♯] Center for Computational Mathematics, Flatiron Institute
[‡] Atomwise
{hytruongson,risi}@uchicago.edu
brandona@jfi.uchicago.edu

## 1 Architecture

As discussed in the main text, our Cormorant architecture is constructed from three basic building blocks: (1) an input featurization that takes $(Z_i, \mathbf{r}_i)$ and outputs a scalar, (2) a set of covariant CG layers that update $F_i^s$ to $F_i^{s+1}$, (3) a layer that takes the set of covariant activations $F_i^s$, and construct a permutation and rotation invariant regression target.

### 1.1 Notation

Throughout this section, we will follow the use the main text, and denote a SO(3)-vector at layer $s$ by $F^s = (F_0^s, \dots, F_L^s)$ with maximum weight $L$. Each SO(3)-vector has corresponding type $\tau^s$, and lives in a representation space $F^s \in V^s = \bigoplus_{\ell=0}^{L^s} \bar{V}_\ell^{\tau_\ell^s}$, where $\bar{V}_\ell = \mathbb{C}^{(2\ell+1)\times 1}$ is the representation space for irreducible representation of SO(3) with multiplicity 1. We will also introduce the vector space for the edge network $V_{\text{edge}}^s = \bigoplus_{\ell=0}^{L^s} \mathbb{C}^{\tau_\ell^s}$.

See Table 1 for a more complete table of symbols used in the supplement and main text.

### 1.2 Overall structure

The Cormorant network is a function $\text{CORMORANT}\left(\{Z_i, \mathbf{r}_i\}\right) : \mathbb{Z}^N \times \mathbb{R}^{N\times 3} \to \mathbb{R}$ that takes a set of $N$ charge-positions $\{Z_i, \mathbf{r}_i\}$ and outputs a single regression target. The

$$\text{CORMORANT}\left(\{Z_i, \mathbf{r}_i\}\right) = \text{OUTPUT}\left(\text{CGNet}\left(\text{INPUT}\left(\{Z_i, \mathbf{r}_i\}\right)\right)\right) \tag{S1}$$

networks are constructed from three basic units:

1. $\text{INPUT}\left(\{Z_i, \mathbf{r}_i\}\right) : \mathbb{Z}^N \times \mathbb{R}^{N\times 3} \to (\bar{V}_0)^N$ which takes the $N$ charge-position pairs and outputs $N$ sets of scalar feature vectors $c_{\text{in}}$. (See section 1.3.)

2. $\text{CGNet}\left(\{F_i, \mathbf{r}_i\}\right) : (\bar{V}_0)^N \times \mathbb{R}^{N\times 3} \to \bigoplus_{s=0}^S (V^s)^N$ takes the set of scalar features from $\text{INPUT}\left(\{Z_i, \mathbf{r}_i\}\right)$, along with the set of positions for each atom, and outputs a SO(3)-vector for each level $s = 0, \dots, S$ using Clebsch-Gordan operations. (See section 1.4.)

3. $\text{OUTPUT}\left(\bigoplus_{s=0}^S (V^s)^N\right) \to \mathbb{R}$ takes the output of CGNet above, constructs a set of scalars, and then constructs a permutation-invariant prediction that can be exploited at the top of the network. (See section 1.5.)

This design is organized in a modular way to separate the input featurization, the covariant SO(3)-vector layers, and the output regression tasks. Importantly, the INPUT and OUTPUT networks are

| Symbol | Meaning |
|---|---|
| $Z_i$ | Atomic charge of atom $i$ (e.g., Hydrogen would be $Z_i = 1$, and Carbon would be $Z_i = 6$) |
| $\mathbf{r}_i$ | Position of atom $i$ |
| $\mathbf{r}_{ij}$ | Relative position of atoms $i$ and $j$: $\mathbf{r}_{ij} = \mathbf{r}_i - \mathbf{r}_j$ |
| $\hat{\mathbf{r}}_{ij}$ | Unit vector in the direction of $\mathbf{r}_{ij}$ |
| $s$ | Index of a CGLayer in Cormorant |
| $S$ | Top CGLayer |
| $\ell$ | Index of an irreducible representation of SO (3) |
| $L$ | Largest index in a representation |
| $m$ | Component $m \in [-\ell, \ell]$ of a SO (3)-vector |
| $\tau^s$ | Vector of multiplicities of a SO (3)-vector at layer $s$ |
| $n_c$ | Number of output channels of a CGLayer |
| $\bar{V}_\ell$ | Representation space for irrep with index $\ell$ and multiplicity $\tau_\ell = 1$: $\bar{V}_\ell = \mathbb{C}^{(2\ell+1) \times 1}$ |
| $V^s$ | Representation space for SO (3)-vector of type $\tau^s$: $V^s = \bigoplus_{\ell=0}^{L} \bar{V}_\ell^{\tau_\ell^s}$ |
| $V_{\text{edge}}^s$ | Vector space for an scalar edge network at layer $s$: $V_{\text{edge}}^s = \bigoplus_{\ell=0}^{L} \mathbb{C}^{\tau_\ell^s}$ |
| $F_i^s$ | SO (3)-vector atom-activation at layer $s$ for atom $i$ |
| $Y^\ell(\hat{\mathbf{r}}_{ij})$ | Spherical harmonics of weight $\ell$ in the direction of $\hat{\mathbf{r}}_{ij}$ |
| $Y_{ij}$ | SO (3)-vector of spherical harmonics connecting atoms $i$ and $j$: $Y_{ij} = \bigoplus_{\ell=0}^{L} Y^\ell(\hat{\mathbf{r}}_{ij})$ |
| $g_{ij}^s$ | Scalar edge activation layer $s$ connecting atoms $i$ and $j$ |
| $d_{ij}^s$ | Scalar matrix of dot products $d_{ij}^s = F_i^s \cdot F_j^s$ |
| $\eta_{ij}^s$ | Radial basis functions between atoms $i$ and $j$ |
| $G_{ij}^s$ | SO (3)-vector edge activation layer $s$ connecting atoms $i$ and $j$ |
| $W_{s,\ell}^{\text{vertex}}$ | Weight mixing matrix for component $\ell$ of atom SO (3)-vector activations at layer $s$ |
| $W_{s,\ell}^{\text{edge}}$ | Weight mixing matrix for component $\ell$ of scalar edge activations at layer $s$ |

Table 1: Table of symbols used in the main text and supplement.

different for GDB9 and MD17. However, the covariant SO(3)-vector layers CGNet were identical in design and hyperparameter choice. We include these designs and choices below.

## 1.3 Input featurization

### 1.3.1 MD-17

For MD-17, the input featurization was determined by taking the tensor product $\tilde{F}_i = \text{onehot}_i \otimes \vec{Z}_i$, where $\text{onehot}_i$ is a one-hot vector determining which of $N_{\text{species}}$ atomic species an atom is, and $\vec{Z}_i = (1, \tilde{Z}_i, \tilde{Z}_i^2)$, where $\tilde{Z}_i = Z_i / Z_{\max}$, and $Z_{\max}$ is the largest charge in the dataset. We then use a single learnable mixing matrix to convert this real vector with $3 \times N_{\text{species}}$ elements to a complex representation $\ell = 0$ and $N_c$ channels (or $\tau_i = (n_c)$.)

We found for MD-17, a complex input featurization network was not significantly beneficial, and that this input parametrization was sufficiently expressive.

### 1.3.2 QM-9

For the dataset QM-9, we used an input featurization based upon message passing neural networks. We start by creating the vector $\tilde{F}_i = \text{onehot}_i \otimes \vec{Z}_i$ as defined in the previous section. Using this, a weighted adjacency matrix is constructed using a mask in the same manner as in the main text: $\mu_{ij} = \sigma((r_{\text{cut}} - r_{ij})/w)$, with learnable cutoffs/width $r_{\text{cut}}/w$ and $\sigma(x) = 1/(1 + \exp(-x))$. This mask is used to aggregate neighbors $\tilde{F}_i^{\text{agg}} = \sum_j \mu_{ij} \tilde{F}_j$. The result is concatenated with $\tilde{F}$, and passed through a MLP with a single hidden layer with 256 neurons and ReLU activations with an output real vector of length $2 \times n_c$. This is then resized to form a complex SO(3)-vector composed of a single irrep of type $\tau_i = (n_c)$.

### 1.4 Covariant $SO(3)$-vector layers

For both datasets, the central covariant $SO(3)$-vector layers of our Cormorant are identical. In both cases, we used $S = 4$ layers with $L = 3$, followed by a single $SO(3)$-vector layer with $L = 0$. The number of channels of the input tensors at each level is fixed to $n_c = 16$, and similarly the set of weights $W$ reduce the number of channels of each irreducible representation back to $n_c = 16$.

#### 1.4.1 Overview

The algorithm can be implemented as iterating over the function

$$\text{CGLayer}\left(g_{ij}^s, F_i^s, \mathbf{r}_i, \right) : (V_{\text{edge}}^s)^{N \times N} \times \mathbb{R}^{N \times N \times 3} \times (V^s)^N \to (V_{\text{edge}}^{s+1})^{N \times N} \times (V^{s+1})^N$$

where $g_{ij}^s \in (V_{\text{edge}}^s)^{N \times N}$ and is an edge network at level $s$ with $c_s$ channels for each $\ell \in [0, L]$, and $F_i^s \in (V^s)^N$ is an atom-state vector that lives in the representation space at level $s$.

The function $\left(g_{ij}^{s+1}, F_i^{s+1}\right) \leftarrow \text{CGLayer}\left(g_{ij}^s, F_i^s, \mathbf{r}_i\right)$ is itself constructed in the following way:

- $g_{ij}^{s+1} \leftarrow \text{EdgeNetwork}\left(g_{ij}^s, \mathbf{r}_{ij}, F_i^s\right)$
- $G_{ij}^{s+1} \leftarrow \text{Edge2Vertex}\left(g_{ij}^{s+1}, Y^\ell\left(\hat{\mathbf{r}}_{ij}\right)\right)$
- $F_i^{s+1} \leftarrow \text{VertexNetwork}\left(F_{ij}^{s+1}, F_i^s\right)$

where:

1. $\text{EdgeNetwork}\left(g_{ij}^s, \mathbf{r}_{ij}, F_i^s\right) : (V_{\text{edge}}^s)^{N \times N} \times \mathbb{R}^{N \times N \times 3} \times (V^s)^N \to (V_{\text{edge}}^{s+1})^{N \times N}$ is a pair/edge network that combined the input pair matrix $g_{ij}^s$ at level $s$, with a position network $F_{ij,c} = F_c\left(|\mathbf{r}_{ij}|\right)$, and $d_{ij} \sim F_i \cdot F_j$ is a matrix of dot products, all of which will be defined below. This output is then used to construct a set of representations that will be used as the input to the VertexNetwork function below.

2. $\text{Edge2Vertex}\left(g_{ij}^{s+1}, Y_{ij}\right) : (V_{\text{edge}}^s)^{N \times N} \times (V)^{N \times N} \to (V^s)^{N \times N}$ takes the product of the scalar pair network $g_{ij}^{s+1}$, with the $SO(3)$-vector of spherical harmonics $Y_{ij} = \bigoplus_{\ell=0}^L Y^\ell\left(\hat{\mathbf{r}}_{ij}\right)$, to produce a $SO(3)$-vector of edge scalar representations that will be considered in the aggregation step in VertexNetwork.

3. $\text{VertexNetwork}\left(G_{ij}^{s+1}, F_i^s\right) : (V^s)^{N \times N} \times (V^s)^N \to (V^{s+1})^N$ updates the vertex SO(3)-vector activations by combining a "Clebsch-Gordan aggregation", a CG non-linearity, a skip connection, and a linear mixing layer.

#### 1.4.2 Edge networks

Our edge network is an extension of the "edge networks" in Message Passing Neural Networks Gilmer et al. [2017]. The EdgeNetwork function takes three different types of pair features, concatenates them, and then mixes them. We express write the edge network (Eq. (9)) in the main text) with all indices explicitly included:

$$g_{\ell c,ij}^{s+1} = m_{c,ij}^s \odot \sum_{c'} \left( \bigoplus_{c_1} g_{\ell c_1,ij}^s \oplus \bigoplus_{c_2} d_{c_2,ij}^s \oplus \bigoplus_{c_3} \eta_{\ell c_3,ij} \right)_{c'} \left( W_{s,\ell}^{\text{edge}} \right)_{c'c} \tag{S2}$$

where:

- $W_{s,\ell}^{\text{edge}}$ is a weight matrix at layer $s$ for each $\ell$ of the edge network.

- $g_{\ell c_1,ij}^s$ is a set of edge activations from the previous layer.

- $d_{c_2,ij}^s = \bigoplus_{\ell=0}^L F_{\ell c_2 i}^s \cdot F_{\ell c_2 j}^s$, is a matrix of dot products, where $F_{\ell c i} \cdot F_{\ell c j} = \sum_m (-1)^m \left(F_{\ell c i,m} F_{\ell c j,-m}\right)$.[1]

- $\eta^s_{\ell c_3, ij} = \eta^s_{\ell c_3}\left(|\mathbf{r}_{ij}|\right)$ is a set of learnable basis functions. These functions are of the form $\eta^s_{\ell c_{k,n}}(r) = r^{-k}\left(\sin\left(2\pi\kappa^s_{\ell n}r + \phi^s_{\ell n}\right) + \mathrm{i}\sin\left(2\pi\bar{\kappa}^s_{\ell n}r + \bar{\phi}^s_{\ell n}\right)\right)$, where $\kappa^s_{\ell n}$, $\bar{\kappa}^s_{\ell n}$, $\phi^s_{\ell n}$, and $\bar{\phi}^s_{\ell n}$ are learnable parameters, the list of channels $c$ is found by flattening the matrix indexed by $c_3 = (k, n)$, and $\mathrm{i}^2 = -1$.

- $\mu^s_{\ell c, ij}$ is a mask that is used drop the radial functions smoothly to zero. This mask is constructed through
$$\mu_{c,ij} = \sigma\left(-\left(r_{ij} - r^s_{c,\text{soft}}\right)/w^s_c\right),$$
where $\sigma(x)$ is the sigmoid activation, $r^s_{c,\text{soft}}$ is a soft cutoff that drops off with width $w^s_c$.

### 1.4.3 From edge scalar representations to $SO(3)$-vector

The function $G^{s+1}_{ij} \leftarrow \text{Edge2Vertex}\left(g^{s+1}_{ij}, Y^\ell\left(\hat{\mathbf{r}}_{ij}\right)\right)$ will take the scalar output of the edge network $g^{s+1}_{ij}$, and construct a set of $SO(3)$-vector representations using spherical harmonics through:

$$G^{s+1}_{\ell c,ij} = g^s_{\ell c,ij}Y^\ell\left(\hat{\mathbf{r}}_{ij}\right) \tag{S3}$$

We note the normalization of the spherical harmonics here is not using the "quantum mechanical" convention, but rather are normalized such that $\sum_m \left|Y^\ell_m\left(\hat{\mathbf{r}}\right)\right|^2 = 1$. This is equivalent to scaling the QM version by $Y^\ell_m\left(\hat{\mathbf{r}}\right) \rightarrow \sqrt{\frac{2\ell+1}{4\pi}} \times Y^\ell_m\left(\hat{\mathbf{r}}\right)$.

### 1.4.4 Vertex networks

The function VertexNetwork is found by concatenating three operations:

$$F^{s+1}_{\ell,i} = \left(\text{VertexNetwork}\left(G^{s+1}_{ij}, F^s_i\right)\right)_\ell \tag{S4}$$

$$= \sum_{c'}\left(\bigoplus_{c_1} F^{s+1,\text{ag}}_{c_1,i} \oplus \bigoplus_{c_2} F^{s+1,\text{nl}}_{c_2,i} \oplus \bigoplus_{c_3} F^{s+1,\text{id}}_{c_3,i}\right)_{\ell,c'}\left(W^{\text{vertex}}_{s,\ell}\right)_{c'c} \tag{S5}$$

where

1. $F^{s+1,\text{ag}}_i = \sum_{j \in N(i)} G^{s+1}_{ij} \otimes_{\text{cg}} F^s_j$ is a CG-aggregation step and $G^s_{ij}$ is the set of edge representations calculated by Edge2Vertex.
2. $F^{s+1,\text{nl}}_i = F^s_i \otimes_{\text{cg}} F^s_i$ is a CG non-linearity.
3. $F^{s+1,\text{id}}_i = F^s_i$ is just the identity function, or equivalently a skip connection.
4. $W^{\text{vertex}}_{s,\ell}$ is a atom feature mixing matrix.

## 1.5 Output featurization

The output featurization of the network starts with the construction of a set of scalar invariants from the set of activations $F^s_i$ for all atoms $i$ and all levels $s = 0\ldots S$. We extract three scalar invariants from each activation $F$ (dropping the $i$ and $s$ indices):

1. Take the $\ell = 0$ component: $\xi_0(F) = F^s_{\ell=0}$.
2. Take the scalar product with itself: $\xi_1(F) = \text{Re}[\tilde{\xi}_1(F)] + \text{Im}[\tilde{\xi}_1(F)]$ where $\tilde{\xi}_1(F) = \sum_{m=-\ell}^\ell (-1)^m F^s_{\ell,m} F^s_{\ell,-m}$.
3. Calculate the SO(3)-invariant norm: $\xi_2(F^s) = \sum_{m=-\ell}^\ell F^s_{\ell,m}\left(F^s_{\ell,m}\right)^*$.

These are then concatenated together to get a final set of scalars: $x_i = \bigoplus_{s=0}^S \xi_0(F^s_i) \oplus \bigoplus_{\ell=0}^L (\xi_1(F^s_i) \oplus \xi_2(F^s_i))$ and fed into the output network network.

### 1.5.1 MD-17

The output for the MD-17 network is straightforward. The scalars $x_i$ are summed over, and then a single linear layer is applied: $y = A\left(\sum_i x_i\right) + b$.

### 1.5.2 QM-9

The output for the QM-9 is constructed using two multi-layer perceptrons (MLPs). First, a MLP is applied to the scalar representation $x_i$ at each site. The result is summed over all sites, forming a single permutation invariant representation of the molecule. This representation is then used to predict a single number used as the regression target: $y = \mathrm{MLP}_2\left(\sum_i \mathrm{MLP}_1(x_i)\right)$. Here, both $\mathrm{MLP}_1$ and $\mathrm{MLP}_2$ have a single hidden layer of size 256, and the intermediate representation has 96 neurons.

### 1.6 Weight initialization

All CG weights $W^\ell$ were initialized uniformly in the interval $[-1, 1]$, and then scaled by a factor of $W^\ell_{\tau^{\mathrm{in}}_\ell, \tau^{\mathrm{out}}_\ell} \sim \mathrm{Unif}(-1, 1) * g/(\tau^{\mathrm{in}}_\ell + \tau^{\mathrm{out}}_\ell)$, where $\tau^{\mathrm{in}}_\ell$, $\tau^{\mathrm{out}}_\ell$ and $g$ is the weight gain.

We chose the gain to ensure that the activations at each level were order unity when the network is initialized. We found that if the gain was too low, the CG products in higher levels would not significantly contribute to training, and information would only flow through linear (one-body) operations. This would result in convergence to poor training error. On the other hand, if the gain is set too high, the CG non-linearities dominate at initialization and would increase the change of the instabilities discussed above.

In practice, the gain was hand-tuned by such that the mean of the absolute value of the CG activations $1/(N_{\mathrm{atom}} N_{\mathrm{c}} (2\ell + 1)) \sum_{\ell, i, c, m} |F^s_{\ell, i, c, m}|$ at each level was approximately unity for a random mini-batch. For experimental results presented here, we used a gain of $g = 5$.

## 2 Experimental details

We trained our network using the AMSGrad [Reddi et al., 2018] optimizer with a constant learning rate of $5 \times 10^{-4}$ and a mini-batch size of 25. We trained for 512 and 256 epoch respectively for MD-17 and QM-9. For each molecule in MD-17, we uniformly sampled 50k/10k/10k data points in the training/validation/test splits respectively. In QM-9 the dataset was randomly split to 100k molecules in the train set, with $10\%$ in the test set, and the remaining in the validation set. We removed the 3054 molecules that failed consistency requirements [Ramakrishnan et al., 2014], and also subtracted the thermochemical energy [Gilmer et al., 2017] for the targets $C_v$, $U_0$, $U$, $G$, $H$, ZPVE.

For both datasets, we used, $S = 4$ CGLayers with $L = 3$ and we used $N_c = 16$ channels at the output of each CGLayer. This gave networks with 299808 and 154241 parameter respectively for QM-9 and MD-17. Training time for QM-9 is takes roughly 48 hours on a NVidia 2080 Ti GPU. Training time for MD-17 varies based upon the molecule being trained, but typically ranges between 26 and 30 hours.

### 2.1 Training instabilities

Training our Cormorant had several subtleties that we both believe are related to the nature of the CG non-linearity. We found a poor choice of weight initialization or optimization algorithm will frequently result in either: (1) an instability resulting in very large training loss ($> 10^6$), from which the network will never recover, or (2) convergence to weights where the activation of CG non-linearities in higher layers turn off, and the resulting training error is poor.

We believe these difficulties are a result of the CG non-linearity, which is quadratic and unbounded. In fact, our network is just a high-order polynomial function of learnable parameters.[2] For the hyperparameters used in our experiments, the prediction at the top is a sixteenth order polynomial of our network's parameters. As a result, in certain regions of parameter space small gradient updates can result in rapid growth of the output amplitude or a rapid drop in the importance of some channels.

These issues were more significant when we used Adam [Kingma and Ba, 2015] then AMSGrad, and when the network's parameters were not initialized in a narrow range. Using the weight initialization

scheme discussed in Sec. 1.6, we were able to consistently converge to low training and validation error, provided we were limited to at most four CG layers.

## Footnotes

[1]Note that $F_{\ell c i} \cdot F_{\ell c j} = \sum_m (-1)^m \left(F_{\ell c i,m} F_{\ell c j,-m}\right)$ is (up to a constant) just the CG decomposition $C_{\ell\ell 0}\left(F_{\ell c i} \otimes F_{\ell c j}\right)$. The specific matrix elements of the CG coefficients $C_{\ell\ell}$ are $\langle \ell m_1 \ell m_2 | 00 \rangle \propto (-1)^{m_1} \delta_{m_1,-m_2}$.

[2]This is true for MD-17, although for QM-9, the presence of non-linearities in the fully-connected MLPs adds a more conventional non-linearity.