[Reviews · NeurIPS 2019]

Reviewer 1



The paper is well-written and clearly draws the connection between physical interactions, tensors and the proposed neural network Cormorant. The proposed network is related to earlier work on tensor field networks [Thomas et al] and covariant compositional networks, but presents architectural changes that lead to improved results on the QM9 and MD-17 benchmarks. Confusingly, while the introduction motivates the work by prediction of atomic force fields, only scalar values are predicted in the experiments. This is also part of the definition of Cormorant: "C3. The type of each output neuron is [...] a scalar". This seems not to be compatible with force field predictions, and also some other important chemical properties are vectors (e.g. molecular dipole moment) or tensors (e.g. polarizability tensor). Would a prediction of these properties also be possible with the current architecture? Section 2 describes between physical interactions in molecules as a motivation for the network architecture. However, electrostatic interactions given here are less important for bonded interaction, but related to long-range and intermolecular interactions. On the other hand, the experiments cover only predictions of small, single molecules. Beyond that, the dipole/dipole interactions given here can also be written as interactions purely defined on charges and atom positions. It would be nice if the authors would give an intuition, why it still makes sense to include these interactions explicitly in the network. Finally, the authors write in the conclusion, that "Cormorant is the first neural network architecture in which operations implemented by the neurons is directly motivated by the form of known physical interactions" . Besides the point that these are not the correct interactions for bonded atoms, there is indeed previous work on this, e.g. Morawietz et al J. Chem. Phys 136, 2012, Gastegger et al, Chem. Sci. 8 (10), 2017, Yao et al. Chem. Sci. 9(8), 2018. Minor issues: - What is the training prediction time required by the network? It seems to collect quite a large number of high-dimensional features. - Table 1: the right table misses units - l. 233 typo: "of of"

Reviewer 2



In this paper the authors introduce the Cormorant neural network, which is constructed to generate rotationally covariant feature representations. Therefore, it could be used to learn tensorial objects that obey the correct transformation rules. In general this paper fulfills all the requirements to be a NeurIPS paper, there is no doubt of its originality and quality. In terms of clarity, the authors give a clear presentation of the physical motivation in sections 2 and 3. Nevertheless, because of its robust and involved mathematical formulation and its implementation technical difficulties some parts of section 4 (sections 4.4 and 4.4.1) are not clear. Still, the general structure of the paper should stay the same due the insightful presentation of the model in sections 2-3, but a clearer presentation should be given in section 4. I would suggest moving implementation/technical details (sections 4.4 and 4.4.1) to the supplementary material and take the necessary number of pages to provide an understandable explanation. The new space left in the main text could be used to elaborate in the description in sections 4.1, 4.2 and 4.3. This would greatly improve the presentation of the paper. Additionally, the authors should address the next list of comments: Introduction: The authors should add a fundamental pair of references by Alessandro De Vita’s group on covariant learning (Physical Review B 95 (21), 214302, 2017) and the recent work on covariant force learning with NNs by Mailoa et al. 2019 (arXiv:1905.02791). At the end of the introduction the authors state: “Cormorant is arguably one of the most general. ” There are no foundations to this statement. The authors have to provide a convincing argument otherwise they have to remove such statement. Section 2: In line 65: What does “(spin)” stands for? If this is in the physics context (e.g. spin of an electron), it is a vectorial quantity, not an scalar as stated by them in line 78. Section 3: Line 103-104, The authors should explicitly mention what is d. Line 122: Why \hat(Q)_{l,i} instead of just Q_{l,i}? Explain. Line 126-127: Is \bar(T)^{(l)} a tensor or a vector according to Line 100? Just to be clear. Section 3.1: It would be very useful to provide a reference for equation 6 to get a better understanding of the content of this section. Section 4 Lines 166-168: The term “Physical laws” is an extremely general term, which makes the paragraph ambiguous. The authors should refer to conserved physical quantities such as linear momentum, angular momentum and energy conservation corresponding to translation, rotation and time invariance, respectively. Line 191: The authors should list what are the differences respect to known architectures in the field of potential energy surface learning. Since translation and rotational invariance are implemented in the same way as any other NN. Section 4.2 Line 233: “of of”, Line 274: “be be” Section 5: QM9 has 12 learnable variables, why learning only 9? In section 5.QM9, The authors should discuss why in particular the Cormorant performs better or comparably good in those particular quantities and why it underperforms for R^2 and \mu? The MD17 dataset consists in energies, forces and molecular coordinates. The models DeepMD and DTNN were trained using only energy labels, SchNet was trained on energies and forces, and the GDML was trained only on forces. The authors should mention which labels did they use for training. In case that they only used energies, can they comment on the possibility of using also forces for training and what would they expect? In the particular case of the comparison with the MD17 dataset, the authors are contrasting they results to old results (Chmiela et al. 2016), even though there a more resent publication on this dataset (Chmiela et al. Nat Commun. 9, 3887, 2018). Such comparison makes more sense given that those are newer results and still Cormorant performs better. It is noteworthy that Cormorant excels a kernel method in this arena.

Reviewer 3



The manuscript presents a new method for modelling physical interactions in molecules, which builds on recent developments in covariance/equivariant neural networks. It is well-structured, and provides a good introduction to the most common physical interactions, and uses this to provide a clear motivation for their model. Although the paper is partially an application of earlier theoretical results, it is a novel way to apply the theory, and the resulting method, in my opinion, represents one of the most clear demonstrations of the potential benefits obtainable when encoding the relevant symmetries directly into neural architectures. Although I am generally positive about the paper, I have some concerns that should be addressed before the paper can be published. The paper would benefit from making stronger connections to earlier work, in particular for making the methodology more accessible to the community. Since slightly different notation and concepts are used in the different prior work, it would be enlightening if the authors in this paper clearly explained how the the presented network differs from that of the Tensor Field Networks paper by Thomas et al and the SE(3) net paper by Weiler et al. More precisely, section 4.1 and 4.2 should both be concluded with a brief description of these architectures compare to these earlier works. For instance, it seems that the covariant CG layers corresponds fairly closely to these earlier efforts, except the connectivity between nodes - where the current work ensures that contributions are attributed to individual atoms and uses a mask on the edge functions instead of convolutions to encode locality. Such similarities should be clearly stated, and the differences clearly motivated - so that it is easier for the reader to understand how these papers are related. I was a bit confused about the notation in line 209. It would seem that s in the layer index. However, in that case I do not understand that the OUTPUT sum operation in line 212 is over the layers s=0...S rather than the activations, i, in the last layer. Is this a mistake, or are you operating over activations from all the layers at once (in which case it in unclear how you aggregate over i)? It would help the reader if the manuscript contained a figure describing the architecture of the network. "Only at higher layers, when the features F_i are well constructed is it likely that a clear mapping to a physical degrees of freedom be possible". Why would it correspond to a physical degree of freedom? Do you mean that it corresponds to a physical n-body interaction? Line 307: "With the exception of R^2, which is much larger than the competitors". Do the authors have any insights into why this particular target is difficult to predict for your model? The authors could perhaps comment on what the difference is between the QM-9 and MD-17 cases, that might explain the considerably better results obtained on the latter data set. I encourage the authors to remember to upload their source code for the final camera-ready version of the paper, as they stated in their Reproducibility Checklist. In particular since there is a considerable theoretical barrier to entering the field of the equivariant/covariant networks, it would be very helpful of the code was available. Quality: The submission is of high technical quality. However, as mentioned above, the manuscript would benefit from a clearer description of how the methodology connects with earlier work, and a brief reflection on why the approach fails on some of the experiments (and works on others). Clarity: The manuscript is clearly written, and motivated the problem and methodology well. I have a few minor suggestions that could improve the clarity further: 1. line 41 and 409: The "Weiler and Welling" citation includes less than half the authors of the paper, and no description of the venue where it was published. Please check other references as well. Gilmer et al for instance also seems incomplete. 2. line 202 "discuss the generalization of convolution the action of compact groups". Missing "to" after "convolution"? 3. line 210 "possibly a scalar function of relative positions". What does "possibly" refers to? When is this possibility exercised? 4. line 262 "We use the ... serve as a starting point". "use"->"let"? 5. line 275 "followed and projecting". Remove "followed"? 6. line 302 "(See the Supplement for more details, including units.)". I could not immediately find this in the supplement (but the units are already in Table 1). 7. line 323 "by the fact all actions" Missing "that" before "all"? Originality: While the manuscript is partially an application of earlier methodological contributions on equivariant networks, it is definitely an original contribution. As mentioned above, earlier work is cited, but more could be done to highlight similarities and differences to earlier work. Significance: The methodology described is a fundamentally new way of describing molecular interactions, and could have substantial impact within the molecular force field community. The modelling approach itself could find applications more broadly within the ML community. Update after rebuttal: The authors have addressed my concerns. I have updated my score to 8.

[Author Response · NeurIPS 2019]

# AUTHOR RESPONSE TO THE REVIEWS OF "CORMORANT: COVARIANT MOLECULAR NEURAL NETWORKS"

We thank all three reviewers for their insightful comments and positive evaluations of our manuscript. We will update the paper to reflect their suggestions by the camera ready deadline.

We will shortly release a Python library that implements the Cormorant architecture. Currently we are just cleaning up and documenting the code. As for the other points brought up by the reviewers we have the following comments:

1. **Structure and supplement:** As suggested by Reviewer 2, we will move some details of the technical implementation in Section 4.4 to the supplement. This will provide room to more clearly explain some of the "big picture" arguments, as requested by several of the reviewers.

2. **Motivation and physical laws:** Our discussion of electrostatics was only intended to illustrate the algebraic structure of interatomic interactions (of any type) rather than suggest that electrostatics is the *only* type of interaction that we care about. The point of the paper is that we do not need to tell the network what interactions exactly it should learn (it will figure that out by itself), but there are nonetheless physical constraints stemming from symmetries that can be explicitly imposed.

3. **Learning force fields and other vector quantites:** In the present version of the paper we only train on and predict scalar-valued quantities. However, predicting forces in MD-17 would be as simple as taking the gradient of the predicted potential energy. *Learning* from forces requires a loss formulated in terms of the gradient, which can be neatly slotted into modern deep learning frameworks, thanks to automatic differentiation. We just did not reach a point where we could add this by the submission deadline. Regarding predicting vector valued quantities in general, this would only require a slight extension of Cormorant. The reason that in Definition 2 constrain ourselves to scalar outputs is essentially just didactic, to explain the relationship between covariance and invariance.

4. **Citations:** We have cleaned up the bibliography, as requested. We also added the missing references pointed out by the reviewers. Several of these papers we were not familiar with, so we are grateful for the suggestions, especially since they help put our work in a broader context.

5. **Connections with other covariant architectures:** As suggested the reviewers, we will make more explicit the relationship between our work and other covariant architectures, including Tensor Field Networks, SE(3)–covariant networks, and the references suggested by Reviewer 2. Unfortunately, this one page is not enough to properly explain the connections.

6. **"Cormorant is one of the most general architectures":** We can show that Tensor Field Networks for example a special case of a Cormorant-like architecture. However, we agree that without explaining this explicitly our statement sounds a bit boisterous, so we will remove it.

7. **Datasets and experimental results:** We are working on understanding why Cormorant performed relatively poorly on $R^2$ and $\mu$. We will update the manuscript if we have a satisfactory answer.

8. **Why only nine variables for GDB9?** We found that $U$, $G$ and $H$ behave very similarly to $U_0$ so it seemed redundant to include them given that we were squeezed for space (Faber et al. did the same).

9. **Spin:** We do not understand R2's comment about spin. Spin is indeed a vectorial quantity. Line 78 says that the interaction energy between two spins is a scalar, not the spins themselves. Please explain.

10. **Number of parameters:** For MD-17 about 46K, whereas for GDB-9 it is 180K.

11. **Aggregating over all layers on top:** Yes, line 208 implies that at the top of the network we aggregate from all covariant layers. This makes sense for extensive quantities like energy (essentially this is how fast multipole methods work). For other learning targets it is harder to justify, so we are experimenting with the network architecture to see if we can avoid these skip connections.

12. **Minor suggestions and typos:** We found the all suggestions by the reviewers useful, and will incorporate them into the manuscript. We will also fix all the typos.

13. **Reproducibility:** The release of the full Cormorant Pytorch library by the camera ready deadline will ensure that our results are fully reproducible.

[Meta-Review · NeurIPS 2019]

All reviewers agree that this is a refreshing and interesting paper. Acceptance is therefore recommended.